# Small RNAs Asserting Big Roles in Mycobacteria

**DOI:** 10.3390/ncrna7040069

**Published:** 2021-10-29

**Authors:** Fatma S. Coskun, Przemysław Płociński, Nicolai S. C. van Oers

**Affiliations:** 1Departments of Immunology, University of Texas Southwestern Medical Center, Dallas, TX 75390-9093, USA; fatma.coskun@utsouthwestern.edu; 2Department of Microbiology, Biotechnology and Immunology, University of Łódz, Banacha 12/16, 90-237 Łódź, Poland; przemyslaw.plocinski@biol.uni.lodz.pl; 3Departments of Microbiology, Pediatrics, University of Texas Southwestern Medical Center, Dallas, TX 75390-9093, USA

**Keywords:** mycobacteria, small RNAs, sncRNAs, RNA processing

## Abstract

Tuberculosis (TB) is an infectious disease caused by *Mycobacterium tuberculosis* (Mtb), with 10.4 million new cases per year reported in the human population. Recent studies on the Mtb transcriptome have revealed the abundance of noncoding RNAs expressed at various phases of mycobacteria growth, in culture, in infected mammalian cells, and in patients. Among these noncoding RNAs are both small RNAs (sRNAs) between 50 and 350 nts in length and smaller RNAs (sncRNA) < 50 nts. In this review, we provide an up-to-date synopsis of the identification, designation, and function of these Mtb-encoded sRNAs and sncRNAs. The methodological advances including RNA sequencing strategies, small RNA antagonists, and locked nucleic acid sequence-specific RNA probes advancing the studies on these small RNA are described. Initial insights into the regulation of the small RNA expression and putative processing enzymes required for their synthesis and function are discussed. There are many open questions remaining about the biological and pathogenic roles of these small non-coding RNAs, and potential research directions needed to define the role of these mycobacterial noncoding RNAs are summarized.

## 1. Introduction

*Mycobacterium tuberculosis* (Mtb) remains one of the leading infectious causes of human mortality, supplanted only in 2020 by the COVID-19 pandemic triggered by the SARS-CoV-2 virus. Mtb evolved from an ancestral smooth tubercule bacillus (e.g., *M. canettii*, *M. pseudotuberculosis*), acquiring virulence elements to attain its preferred pathogenicity towards humans [1]. The acquisition of these virulence elements coincided with Mtb undergoing a genomic downsizing relative to the 100 different smooth tubercule bacilli species characterized [1,2]. Despite this downsizing, a core genome is evident among the pathogenic strains of mycobacteria. Several decades of research efforts have been devoted to understanding how the ~4000 protein-coding elements evident in the Mtb genome contribute to growth, survival, and pathogenic processes [3,4,5,6,7]. Recent technical advances in deciphering the complex nature of Mtb and related mycobacterial genomes, including improved large-scale RNA-sequencing strategies, have revealed an abundance of small RNAs (sRNA). First, described as ranging in size from 50 to 350 nucleotides (nts) [8,9,10,11,12], these small RNAs now include some as small as 18 nts [13]. The sRNAs, originally selected with sequences > 100 nts in length, were found to represent ~11% of the intergenic transcripts (IGRs) identified from the exponential phase cultures. In addition to the sRNAs, IGRs include 5′ and 3′ UTRs, tRNAs, and antisense RNAs. Based on the normalized read counts for sense, antisense, and intergenic noncoding RNAs, the antisense and intergenic noncoding RNAs made up roughly 25% of the transcripts mapping outside of ribosomal RNA genes [10]. The sRNAs are detected in both exponential and/or stationary phase cultures, in infected eukaryotic cells, and in patients with tuberculosis (TB), suggesting key roles in all aspects of mycobacterial growth and survival [9,10,11,12,13,14]. In the current review, recent discoveries pertaining to these sRNAs are described. Emerging reports detailing their diverse functions along with their transcriptional regulation and processing are discussed. Future directions of research and therapeutic strategies to manipulate such sRNAs are also presented.

## 2. Identification and Designation of Mycobacterial Small RNAs

The initial screens identifying Mtb-encoded sRNAs were prompted by prior reports on the existence of small noncoding RNAs in other bacterial species such as *Escherichia coli*, *Salmonella enterica,* and *Staphylococcus aureus* [15,16,17]. As with most pioneering studies, assorted definitions and naming strategies were applied to the Mtb small RNAs [9,10,11,12,13,18,19]. One group of the sRNAs are called intergenic sRNAs, transcribed from intergenic regions. These are sometimes termed trans-encoded RNAs. Such intergenic sRNAs primarily use a 6–7 nucleotide seed-sequence that targets complementary sequences on genes with imperfect base-pairing. A second group partially or completely overlaps with protein-coding transcripts in either the sense or antisense orientation. These sRNAs often hybridize to the coding RNA with perfect base–pair complementarity. This interaction changes the translation efficiency of the corresponding gene or induces the degradation of the target mRNA [20,21].

The first Mtb-encoded small RNAs identified involved enriching for low molecular weight RNAs from exponential and stationary phase Mtb cultures followed by cDNA synthesis and sequence analysis [9]. Four antisense (ASdes, ASpks, AS1726, and AS1890) and five intergenic sRNAs (B11, B55, C8, F6, and G2), ranging in size from 38 to 238 nts, were identified (Table 1). Later work revealed that ASdes, ASpks, AS1726, and AS1890 were also present in Bacillus Calmette Guerin (BCG) mycobacterial cultures [22]. ASdes was subsequently detected in the plasma of 15 of 27 TB patients and 6 of 24 BCG vaccinated individuals [22,23,24].

Additional RNA sequencing approaches have since broadened the number of small RNAs detected in exponential phase Mtb cultures, with 19 antisense and 20 intergenic sRNAs identified [10]. Of the 20 intergenic sRNAs, 3 have been characterized. Designated as *Mycobacterium tuberculosis* sRNAs (MTSs), MTS2823 was the most abundant sRNA in the log phase cultures. Its levels increased 6.5-fold when the cells reached stationary phase. MTS0997 and MTS1338 increased 1.6-fold and 23.2-fold in stationary relative to exponential phases of mycobacterial growth [10]. Since these first screens, 27 additional antisense sRNAs were reported in dormant TB in addition to the 2 antisense and 8 intergenic sRNAs previously identified [19].

Thirty-seven sRNAs have been identified in *Mycobacterium bovis* (*M. bovis*) cultures, 19 of which were confirmed by Northern blotting (Table 1) [11]. These were given the designation of Mcr1-McR19 for mycobacterium cloned RNAs. Not surprisingly, as *M. bovis* originated from Mtb, Mcr6, Mcr11, and Mcr14 are the same as the aforementioned C8, MTS0997, and F6 sRNAs, respectively. As RNA sequencing screens are now routinely performed, confirmation of the sRNAs originally identified followed by functional studies is ongoing. Mcr3, Mcr4, Mcr8, Mcr9, Mcr11, and Mcr14 are consistently detected in *M. bovis*, while B11, B55, C8, F6, G2, and ASdes are evident in Mtb [8]. Given that many sRNAs, described in other bacterial species, are formed in more extreme growth conditions, distinct clusters of Mtb-encoded sRNA are also found to be induced following either oxidative stress, pH stress, membrane stress, or nutrient or iron starvation conditions [12]. To date, little has been reported about whether these sRNAs are produced following antibiotic treatments in patients. Overall, the number of sRNAs now known is >189, with a new nomenclature provided to distinguish them [32]. Almost all these sRNAs ranged in size from 51 to 350 nts.

While most of the RNA screens with Mtb focused on RNAs > 50 nts, many smaller RNAs < 50 nts have been characterized in non-mycobacterial species [33,34,35,36]. For example, *Salmonella* expresses a small RNA called Sal-1, which is generated from the 5′ end of a ribosomal RNA by the eukaryotic miRNA processing enzymes [33]. Sal-1 targets the inducible nitric oxide synthase, with the pathogenic role for this sRNA established by the increased killing of Sal-1 deficient *Salmonella* in infected epithelial cells [33]. Sal-1 resembles eukaryotic miRNAs, which are small noncoding RNAs (20–22 nts) that use 6–7 nucleotide seed sequences to mediate the degradation of mRNAs [37,38]. The first screen for such miRNA-like sRNAs in mycobacteria was undertaken with *Mycobacterium marinum* [39]. In this screen, a single 23-nt RNA was discovered, with features characteristic of a eukaryotic miRNA including the requisite interaction with the Argonaute protein, part of the eukaryotic RNA-induced silencing complex [39]. To date, the *M. marinum* sRNA has no ascribed functions or targets. In a screen for miRNA-like sRNAs in TB-infected patients, six distinct Mtb-encoded miRNA-like sRNAs were discovered in the serum [40]. While all of these had 22-nt lengths consistent with the size of miRNAs, their extremely high GC content was unusual (86%–100%). In a broader screen for miRNA-like sequences using comprehensive miRNA selection criteria with the annotations from miRbase, Rfam, and Repbase, where plant small RNAs are also considered, a set of 35 smaller RNAs (<50 nts) were identified in Mtb-infected macrophages [13]. Except for one of these sRNAs, most were only detected in infected macrophages, with their levels increasing over a 6-day infection period. Termed smaller noncoding RNAs (sncRNAs), the sizes of these ranged from 18 to 30 nts. The 35 sncRNAs had an average GC content of 50%. In a technical advance to determine the levels of these sncRNAs, a miRNA-based quantitative RT-PCR was developed. This assay incorporates locked-nucleic acid technologies to provide extremely high specificity and selectivity for short RNAs. The expression changes of three of these Mtb-encoded sncRNAs, sncRNA-1, sncRNA-6, and sncRNA-8, were verified with this technique [13].

To summarize, diverse mycobacterial species produce small RNA transcripts ranging in size from 18 to 350 nts. The secondary structure of representative examples of such sRNAs is shown in Figure 1. The mycobacterial sRNAs have diverse sizes and extensive predicted secondary structures that lack commonality. Only a handful of these sRNAs have been functionally characterized. Some are more abundant in infected cell lines and in patients, implying roles in pathogenesis. Scientists are beginning to explore their targets, production and processing requirements, and contributions to pathogenesis. We describe next the current state of knowledge of some of these sRNAs.

## 3. Functional Roles of Mycobacterial sRNAs and sncRNAs

A key step in identifying putative biological roles for the sRNAs relates to what stages in a mycobacterial growth cycle they are expressed [41]. Additional insights have come from the environmental conditions that affect sRNA expression. Among the conditions are oxidative stress, nutrient deprivation, DNA damage, antibiotic exposure, and/or acidic environments, the latter occurring in the phagolysosome formed in macrophages and dendritic cells. Putative functional roles for the numerous sRNAs need also to consider the stability of the sRNA, affected by both the relative GC content and secondary RNA structures. Examples of several better characterized sRNAs are B11/6C, MTS1338/DrrS, Ms1, MTS0097/Mcr11, ncRv11846/MrsI, Mcr7, and sncRNA-1 (Figure 2, Table 1).

sRNA B11 (93 nts), later named 6C owing to its similarity to a small RNA found in other bacterial species, forms two stem-loops via six conserved cytosines [42,43]. Target sequence searches have suggested that B11/6C regulates Mtb transcripts coupled to DNA replication and protein secretion. In mechanistic studies in *M. smegmatis*, 6C was found to interact with two mRNA targets, *panD* and *dnaB* (Figure 2A, Table 1). Moreover, overexpression of 6C inhibited *M. smegmatis* growth. Several groups have used mycobacterial RNA over-expression vectors to further understand how the various sRNAs function. Over-expression of MTS1338 (117 nts) prevents Mtb replication, suggesting it targets key genes needed for mycobacterial growth (Figure 2B, Table 1) [19]. Later named as DosR regulated sRNA (DrrS), MTS1338 is induced by DosR. High levels of MTS2823 (300 nts) also inhibit Mtb growth, with transcriptome analysis using microarrays revealing that many transcripts involved in metabolism are downregulated (Figure 2B, Table 1) [10].

MTS0997 (131 nts), later named Mcr11, upregulates several genes required for Mtb fatty acid production (Figure 2C, Table 1) [30]. This sRNA positively regulates *rv3282*, *fadA3*, and *lipB* translation by binding a 7–11 nucleotide region upstream of the start codon. Supplementing fatty acids in the mycobacterial cultures overrides this regulatory process, revealing a feedback loop to control metabolic functions in Mtb. The regulatory role for sRNAs in Mtb metabolism is also revealed with ncRv11846 (106 nts). An ortholog of the *E. coli* sRNA RhyB, ncRv11846 is termed mycobacterial regulatory sRNA in iron (MrsI) (Figure 2D, Table 1) [12]. ncRv11846/MrsI is expressed following iron starvation [16]. This sRNA contains a six-nucleotide seed sequence that targets and negatively regulate the transcripts *hypF* and *bfrA*, which encode for nonessential iron-containing proteins. This translational roadblock increases the levels of free iron available. There are additional sRNAs identified that are reduced in expression in response to iron starvation [12]. The role of these sRNAs remains an open question.

Among the diverse sncRNAs, sncRNA-1 remains the best characterized (Figure 2E, Table 1) [13]. This non-coding RNA is present in the RD1 pathogenicity locus, in between *esxA* and *espI*. Over-expressing sncRNA-1 alters the Mtb transcriptome, with multiple genes required for fatty acid biogenesis increased in expression. Screening putative targets of sncRNA-1 by seed-sequence complementarity searches reveals two targets of this sRNA, *rv0242c*, and *rv1094*. These encode two proteins involved in the oleic acid biogenesis pathway. Both genes have putative sncRNA-1 binding sites within their 5′ UTRs. Substituting selected nucleotides involved in Watson‐Crick base pairing, either within the 5′ UTR or in the sncRNA seed sequence, eliminated the positive regulation. One novel approach for studying microRNA functions is the use of locked nucleic acid power inhibitors (LNA-PIs). These have modified RNA sequences that prevent their cleavage by RNA processing enzymes. They also have chemical modifications to enable uptake into cells without any transfection or liposome-based carrier needs [44]. They hybridize with target miRNAs with extremely high specificity, antagonizing the function of the miRNA. These LNA-PIs were tested in mycobacteria, which are inherently difficult to electroporate or transfect with liposome-based technologies. Notably, such LNAs are easily incorporated into mycobacteria and can antagonize sncRNAs in Mtb [13,45]. Incubation of Mtb with an LNA-PI selectively targeting sncRNA-1 abolished the upregulation of the *rv0242c* [13]. This LNA treatment reduced Mtb survival in infected macrophages, revealing a key pathogenic contribution of this sncRNA. The functions of sncRNA-6 and sncRNA-8 remain unexplored [13].

MTS2823 is termed Ms1 as it was functionally characterized in *M. smegmatis* and has homology to the 6S sRNA [26]. Best defined in *E. coli*, 6S sRNA has a secondary RNA structure that resembles an open promoter. The sigma factor bound RNA polymerase (RNAP) holoenzyme has a high affinity for this RNA structure [46]. The 6S sRNA complexes the RNAP, competitively reducing transcriptional activity [46]. Studies in *M. smegmatis* suggest that Ms1 competes with the sigma factor for binding to RNAP, hence suppressing transcriptional activity. Given the complexity of defining RNA‐protein complexes, a revised model is proposed in which Ms1 sequesters the RNAP (Figure 2F, Table 1) [27]. Another negative regulatory sRNA that has been characterized is Mcr7 [31]. This sRNA interferes with the translation of *tatC* mRNA, which encodes twin arginine translocation C (TatC) (Figure 2G, Table 1). TatC is a part of a protein export pathway that is also involved in Mtb pathogenesis [47]. All told, accumulating findings reveal a critical role for sRNAs and sncRNAs in Mtb pathogenicity.

## 4. Regulation of Mycobacterial sRNAs’/sncRNAs’ Expression

As more sRNAs/sncRNAs are discovered in mycobacteria, regulatory elements controlling their expression and processing are slowly being identified and characterized. This includes the identification of key cis- and trans-regulatory factors. In mycobacteria, sigA is the primary transcription factor, which is a member of the sigma70 family [48]. SigA recognizes the consensus cis regulatory sequence, the TTGCGA–N_18_–TANNNT hexamer that is present at −35 and −10 region upstream of the transcription start site (Figure 3A) [49,50]. SigA binding enables RNA polymerase to transcribe at promoter sites responsible for the expression of housekeeping regulons and for mycobacterial growth [48,51]. Miotto et al. developed computational predictions to identify sigA-regulated sRNAs [52]. Of the sRNAs identified in the screen, 46.9% had the consensus SigA promoter sequence in the upstream of the 5′ end, with 8.5% containing an intrinsic or factor-independent terminator sequence in the downstream or 3′ end. While 13.6% of the genes encoding sRNAs had both 5′ and 3′ motifs, their presence and impact on transcription requires further study. The remaining 31.0% of the sRNA encoding genes had neither defined motif, suggesting the involvement of other regulatory factors. For example, the gene encoding Ms1 contains a −10 element, starting five-nucleotide upstream of +1 position along with a distinct −35 element, suggesting that a distinct sigma factor regulates its expression (Figure 3B). Ms1 contains different regulatory elements (−491/+9 region) that contribute to its expression [27].

Coupled with the cis-regulatory elements, novel trans-regulatory elements are being identified that control sRNAs/sncRNA expression. Among these are alternate transcription factors or sigma factors. For instance, sRNA ncRv11846/MrsI has an IdeR binding site in its promoter region. IdeR is an iron-responsive master regulator of genes coupled to iron metabolism, including the sRNA ncRv11846/MrsI (Figure 3C) [12,53]. Mcr7 expression is regulated by PhoP, which is a part of the two-component system PhoP/PhoR [31,54]. Direct binding assays with chromatin immuno-precipitation of PhoP revealed that it binds to the promoter region of Mcr7 to induce its expression in exponential phase Mtb cultures (Figure 3D). sRNA MTS0997/Mcr11 resides between two protein-coding genes *rv1264* and *rv1265*, with the protein products of these two genes involved in the metabolism of cAMP [14]. *rv1264* encodes an adenylyl cyclase, which catalyzes ATP to cAMP. *rv1265* is a transcription factor that binds to both ATP and DNA. DNA binding studies have shown that *rv1265* induces MTS0997/Mcr11 expression (Figure 3E). *rv1265* is now termed AbmR for ATP binding Mcr11 regulator. Mapping studies of the 5′ end of MTS0997/Mcr11 revealed that its −35 element coincides with the promoter regions of AmbR, which is oriented in the opposite direction (Figure 3E) [10]. MTS1338/DrrS is also transcribed in the opposite direction to its neighboring gene called *rv1733c*, but mapping of the TSS of *rv1733c* revealed that it is separated by 190 nucleotides from the TSS of MTS1338/DrrS [55]. *rv1733c* encodes a protein involved in cell wall biogenesis and is a component of the DosR regulon [10]. The DosR regulon, induced by nitric oxide (NO), is the primary mediator of the hypoxic stress response [56]. MTS1338/DrrS is also upregulated in response to NO, and the MTS1338/DrrS promoter is activated by DosR, established with b-galactosidase reporter assays (Figure 3F) [28].

In summary, identification of the cis- and trans-acting factors is revealing many diverse types or regulatory elements involved in the sRNA expression. Little is known about the regulation of the sncRNAs.

## 5. Processing of Mycobacterial sRNAs and sncRNAs

Many sRNAs are generated as full-length mature transcripts with no obvious processing steps. Yet, several of the smaller species do undergo some form of processing from larger single-stranded (ssRNA) precursors [27,28,30]. Among these are Ms1, MTS0997/Mcr11, MTS1338/DrrS, sncRNA-1, and sncRNA-6. Ms1 is a 300 nt transcript detected in both exponential and stationary phase cultures. Notably, it also exists as a 250 nt transcript in stationary phase, suggesting some form of processing [10]. MTS1338/DrrS is transcribed as a precursor transcript of >400 nts (referred to as DrrS+) that is cleaved at the 3′ end to yield the mature 108 nts form [28]. MTS0997/Mcr11 has a 3′ end that varies in size by 3–14 nts, implying that a 3′ RNA processing occurs like that for MTS133/DrrS [30].

Both sncRNA-1 and sncRNA-6, which have final sizes of 25 nts and 21 nts, respectively, require processing enzymes for their generation [13]. These sncRNAs were predicted to exist as precursor transcripts >115 nts that have defined RNA structures involving double-stranded RNAs (dsRNA) segments that form hairpin loops. To identify the putative processing requirements needed for the generation of sncRNA-1, nucleotide substitutions were created within the hairpin loop and antisense complementarity strand of the precursor form of this sncRNA. This caused the formation of multiple intermediate size-transcripts (40–115 nts), detected by Northern blotting [13]. Thus, the processing of the longer RNA transcript depends on both the formation of the hairpin loop and the specific nucleotides at a putative cleavage site needed to form sncRNA-1 [13]. Notably, the expression of the precursor sncRNA-1 transcript, containing sncRNA-1 that was no longer processed into the 25 nt species because of the introduction of nucleotide substitutions, was unable to regulate gene expression. SncRNA-6 also undergoes a sequence-specific processing from a longer RNA transcript. Like sncRNA-1, mutations that disrupt the hairpin loop in which sncRNA-6 resides or the mutations at the cleavage site of sncRNA-6 prevent its processing. Taken together, multiple experiments establish the existence of a small RNA processing system in mycobacteria. These findings do not exclude the possibility that some of the Mtb sRNAs could be generated by miRNA processing enzymes when the mycobacteria are propagating in eukaryotic cells during infections [57].

Several candidate RNA processing enzymes have been reported to date. Among these are ribonuclease E (RNase E), polynucleotide phosphorylase (PNPase or GpsI), ribonuclease J (RNase J), and the ATP-dependent RNA helicase RhlE (Figure 4) [58]. All are components of the RNA degradosome. Except RhlE, all are essential for in vitro growth, determined by identifying key genes through a transposon mutagenesis screen (Himar1 transposon libraries) [7]. Mechanistically, RNase E recognizes the 5′ phosphate of the transcript and then cuts at an A/U rich sequence of the ssRNA [59]. PNPase and RNase J are 3′ and 5′ specific exonucleases, respectively, that stop upon the presence of a dsRNA sequence [60]. Many research teams have made use of CRISPR interference mediated knock down of the RNA processing enzymes to study their role in the generation of specific sRNAs [27,58,61]. Sikova et al. has investigated the contribution of the core RNase enzymes in the processing of Ms1 [27]. Knockdown of PNPase increased the levels of Ms1 ~30%, while the targeting of RNase E and RNase J had no effect on this sRNA, revealing some target specificity. These findings further suggest that the processing of Ms1 likely involves additional RNA processing enzymes. Another possibility is that residual protein levels of PNPase were still resulting in some processing of the longer RNA transcript. Taken together, the limited number of studies on the RNA processing enzymes leave open many questions about how Mtb produces sRNAs from longer transcripts.

## 6. tRNA Processing Enzymes as Potential Players for sRNA Maturation

Transfer RNAs (tRNAs) share some common features with small RNAs, being relatively short and highly structured non-coding RNA molecules. tRNA maturation involves several steps, with both 3′ and 5′ ends being extensively processed in an orchestrated, sequential order. Besides the core RNA degradosome components, tRNA processing enzymes likely play roles in maturation and turnover of certain sRNA species. In eukaryotes, many RNP complexes involved in tRNA biology participate in the generation and subcellular trafficking of other small structured RNAs such as snRNA, snoRNA, 5S RNA, and others [62]. In many organisms, transcripts encoding tRNAs are also a source of regulatory small RNAs, namely tRNA-derived small RNAs [63]. The mechanisms of tRNA maturation in Mtb are not well characterized and require future studies. Compared with *E. coli* and *B. subtilis*, used as model bacteria for RNA processing, Mtb encodes for RNase P, which is involved in the initial processing of the 5′ end of tRNA molecules [64]. The suite of 3′ end processing enzymes includes RNase PH, RNase Z, the oligoribonuclease [65], RNase D (*rv2681*), and a divergent functional and structural ortholog of RNase T (*rv2179c*) [66]. As a large proportion of mycobacterial tRNAs require an enzymatic addition of the CCA sequence at their 3′ end, they are likely additionally processed by the Poly(A) polymerase and/or PNPase. All the ribonucleases described above have the potential to be involved in the processing of small non-coding RNAs other than tRNA. In *E. coli*, RNase PH is implicated in degradation of structured RNAs, which accumulate in the mutant lacking this RNase [67]. In the same model organism, the 3′ exoribonucleolytic trimming is required for the final maturation of multiple small, stable RNA species, and this is carried out primarily by the RNase PH and RNase T [68].

tRNA cleavage is seen in all kingdoms of life as a regulatory mechanism, adding another layer to the complexity of gene regulation mechanisms. This has been observed in *Streptomyces coelicolor*, another actinomycetes related to Mycobacteria [69]. tRNA cleavage has recently been reported in Mtb [70]. Tuberculosis encodes numerous toxin-antitoxin systems, with many requiring a ribonuclease component. The VapC11 ribonuclease of the virulence-associated TA system, VapBC, specifically cleaves two tRNA species, tRNA^Gln32-CUG^ and tRNA^Leu3-CAG^ [70]. Mtb encodes for about 50 VapC ribonuclease toxins, with these having the potential to directly target noncoding RNAs. It is also likely that these would cleave tRNAs to yield tRNA derived functional sRNAs in Mtb, as they seem to have in higher eukaryotes [71]. Direct or indirect interplay between VapC toxins and sRNA is inevitable. In fact, overexpression of the MTS2823 restricts expression of at least five VapC homologues [10], but the exact mechanism remains unexplored.

## 7. The Hunt for the Mycobacterial Hfq Equivalent

In the majority of bacteria species, trans-encoded sRNAs require RNA chaperones, either Hfq or ProQ, to ensure appropriate sRNA/mRNA base pairing [72]. Taking advantage of the 6C sRNA inhibiting *M. smegmatis* growth when overexpressed, a screen for RNA chaperons that mediate the interaction between 6C, and its targets was developed [32]. In experiments where *M. smegmatis* clones overexpressing 6C were exposed to saturation mutagenesis, there was some growth. However, the few colonies that recovered were those that had mutations in the overexpression cassette, indicating the lethality of 6C. No other genetic mutations were observed in the colonies from the saturation mutagenesis library. Presuming a chaperon was targeted, the growth inhibition by B11/6C was not overcome, suggesting a chaperone protein was not involved [43]. It remains possible, however, that the chaperone protein had other essential functions, leaving open a role for as yet unidentified chaperons.

CsrA, a conserved small RNA binding protein, was recently shown to assist in a complex between the sRNA and its mRNA targets in *Bacillus subtilis* [73]. The RNA chaperones closely cooperate and interact with the core RNA degradosome to ensure efficient regulation of gene expression [74]. However, experiments to identify the orthologues of Hfq, ProQ, or CsrA chaperones in Mtb have failed, which suggested that Mtb must exploit alternative proteins or mechanisms for efficient sRNA/mRNA interactions. Such interactions in Mtb were proposed to involve direct Watson‐Crick base pairing involving GC-rich sequences of the sRNA and the target [75]. While this may apply to certain sRNAs, it is likely that most Mtb sRNAs target the mRNAs or DNA through unidentified accessory proteins.

Recent studies from *E. coli* and *S. aureus* have revealed that cold shock domain containing proteins (CSPs), involved in binding and melting RNA species [76,77], also interact with several sRNAs. CSPs typically respond to stress and could aid in a coordinated response to external stimuli that is not limited to cold sensing [78]. Hence, these proteins may likely be involved in sRNA-mediated regulation of gene expression in Mtb. To corroborate this notion, CspA and CspB both associate with the core RNA degradosome in Mtb [58]. Future studies will likely reveal their relevance to the functionalities of the RNA degrading machinery. Interestingly, the mycobacterial CspA gene itself is co-expressed with the sRNA molecule, ncRv3648c. Exploiting active RNA structure unwinding, with the help of ATP-dependent RNA helicases, could theoretically support sRNA folding in the absence of passive unwinding mechanisms provided by Hfq-like chaperones. A previous study from *E. coli* has reported the requirement of the CsdA DEAD-box helicase for low temperature riboregulation of *rpoS* mRNA via sRNA-mediated mechanism, where the activity of Hfq was not sufficient for translational activation of *rpoS* expression [79].

Intriguingly, *M. smegmatis*, *M. dioxanotrophicus*, and *M. goodie* encode a eukaryotic-like protein with a full length TROVE domain (KEGG database search www.genome.jp (accessed on 27 May 2021)), sharing over 35% identity with the human 60 kDa SS-A/Ro ribonucleoprotein ortholog (SIM analysis results [80]). The 60 kDa SS-A/Ro ribonucleoprotein binds to misfolded small RNAs and pre-5S rRNA in eukaryotes [81]. It is thought to function as an RNA chaperone that stabilizes small RNAs of the Y family and protects them from enzymatic degradation [82]. In mycobacteria, the protein coding element was likely acquired by horizontal gene transfer from a mycobacteriophage, with a similar gene identified in the mycobacteriophage Sparky (KEGG database ortholog search, www.genome.jp (accessed on 27 May 2021)). It remains unknown whether the TROVE protein has acquired some functions related to sRNA metabolism in fast growing mycobacterial species or whether it is simply a useless remnant of a previous bacteriophage infection.

## 8. Concluding Remarks

Mtb-encoded small RNAs are emerging as new regulators of mycobacterial growth, survival, and pathogenesis. To date, the functions for only a handful of these sRNAs have been described. Given their distinct sizes, their functional contributions likely differ when comparing those <50 nt and those between 50 and 350 nt. The recently developed CRISPR interference-based assays hold great potential for studying the function of sRNAs and sncRNAs [12,26,61]. In addition, LNA power inhibitors recently validated on sncRNA-1 are a very tractable system for blocking sRNA functions [13]. Another question that remains unexplored is the putative role of such sRNAs in enhancing the mycobacterial resistance to antibiotics. Future studies may explore if these sRNAs are induced in response to antibiotics, which also opens another question on the type and sequence specificity of the regulatory proteins controlling sRNA expression. The lack of a comprehensive study to identify the regulatory factors has limited the identification of additional sRNAs. Lastly, many studies suggest that the assorted sRNAs are processed after transcription, and this adds another complication owing to the lack of elaborate techniques to define exact processing events. The processing may be growth phase or stress-dependent, altering the function of the RNA in selected physiological conditions. Moreover, the mycobacterial proteome involved in RNA processing is still poorly annotated. Using mycobrowser, candidate genes involved in RNA processing in Mtb are evident (Table 2). However, more studies are needed to investigate the role of these putative RNA binding/processing proteins, as most have been identified via computational predictions. As summarized, the last decade has identified many distinct Mtb-encoded RNAs. The next decade will likely address the open questions mentioned throughout this review. Identification of new sRNAs/sncRNAs involved in pathogenesis and their regulatory mechanisms will enhance our understanding of tools that Mtb utilizes to escape macrophage killing, which will eventually help eradicate the TB.

## Figures and Tables

**Figure 1 ncrna-07-00069-f001:**
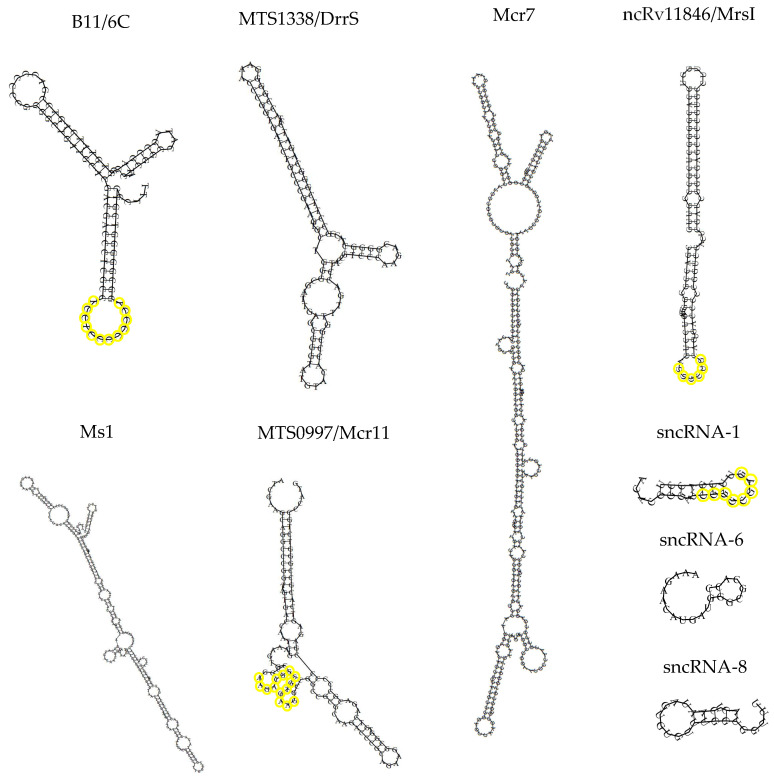
Predicted secondary structure of B11/6C, MTS1338/DrrS, Mcr7, ncRv11846/MrsI, Ms1, MTS0997/Mcr11, sncRNA-1, sncRNA-6, and sncRNA-8. Verified seed sequences are highlighted in yellow. The structures were obtained using the RNAFold web server RNA prediction software. While predicting the structure of ncRv11846/MrsI, 6 nts at the 5′ ends were omitted for simplicity.

**Figure 2 ncrna-07-00069-f002:**
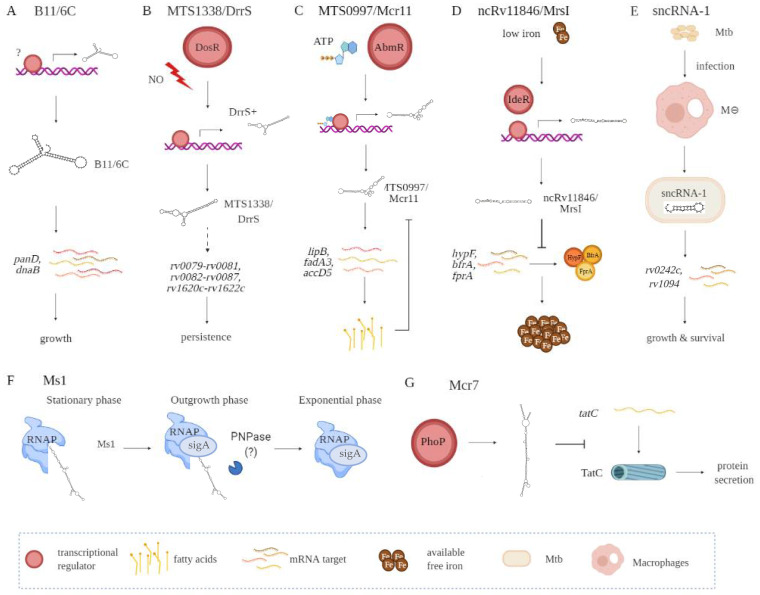
Functional contributions of seve ral Mtb-encoded sRNAs including B11/6C (**A**), MTS1338/DrrS (**B**), MTS0997/Mcr11 (**C**), ncRv11846/MrsI (**D**), sncRNA-1 (**E**), Ms1 (**F**), and Mcr7 (**G**). Indirect interactions are shown with a dashed line. (**A**) B11/6C is induced by undefined factors and positively regulates the expression genes (*panD, dnaB*) coupled to growth. (**B**) DrrS+ is induced by the DosR regulon upon nitric oxide stress. Then, it undergoes post transcriptional processing to yield MTS1338/DrrS. MTS1338/DrrS promotes the expression of three operons (*rv0079-rv0081, rv0082-rv0087,* and *rv1620c-rv1622c*), which cause defects in Mtb growth and promote persistence. The mechanism of this MTS1338/DrrS mediated regulation has not been characterized. (**C**) The expression of MTS0997/Mcr11 is regulated by AbmR, an ATP-bound transcription factor. After transcription, MTS0997/Mcr11 undergoes processing at the 3′ end and then regulates the expression of genes (*lipB, fadA3,* and *accD5*) involved in fatty acid production in a site-specific manner. This is negatively regulated by fatty acids. (**D**) In iron-restricted environments, the iron-responsive transcription factor IdeR induces the expression of ncRv11846/MrsI, which in return hinders the translation of nonessential iron storing proteins (*hypF, bfrA,* and *fprA*). This increases the level of free iron that can be used for essential functions. (**E**) sncRNA-1 is induced in infected macrophages and gets processed to yield 25 nts RNA. The processed sncRNA-1 enhances the expression of *rv0242c* and *rv1094*, two genes involved in oleic acid production. This regulatory network then promotes Mtb growth and survival inside macrophages. (**F**) Ms1 sequesters RNA polymerase (RNAP) at the stationary phase. Upon entrance to the outgrowth phase, Ms1 is degraded by PNPase, and some other RNases not yet identified, which release RNAP to promote global transcription. (**G**) PhoP induces the expression of Mcr7, which abrogates the translation of *tatC*. *tatC* encodes TatC, which is involved in the protein secretion pathway.

**Figure 3 ncrna-07-00069-f003:**
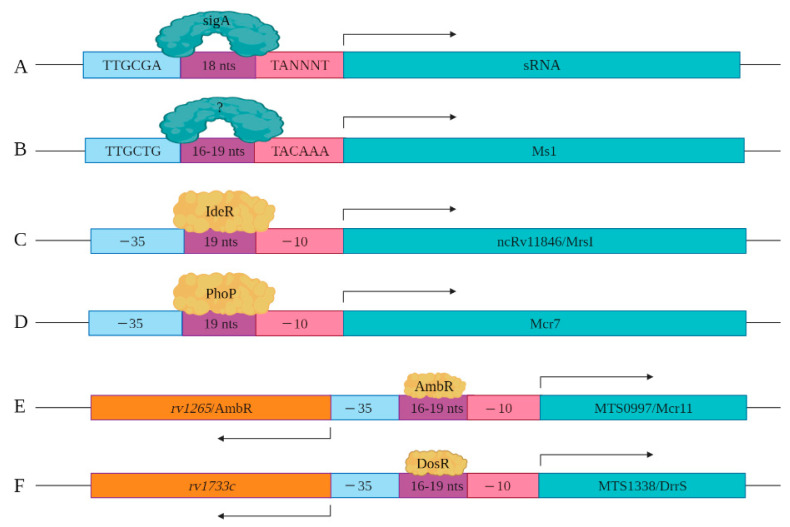
Cis and trans regulatory elements involved in sRNA expression are shown. (**A**) sigA recognizes a consensus sequence to induce the expression of a set of sRNAs identified by Miotto et al. (**B**) −35 and −10 elements upstream of Ms1 are shown. Distal regulatory elements not shown here also contribute to the expression of Ms1. (**C**) IdeR potentially regulates the expression of ncRv11846/MrsI through the IdeR-box found in its promoter region. (**D**) The expression of Mcr7 is regulated by PhoP, which is a part of the PhoPR two-component system. (**E**) The expression of MTS0997/Mcr11 is regulated by AmbR located in the upstream of MTS0997/Mcr11 and expressed in the opposite orientation. (**F**) The expression of MTS1338/DrrS is regulated by the DosR transcription factor.

**Figure 4 ncrna-07-00069-f004:**
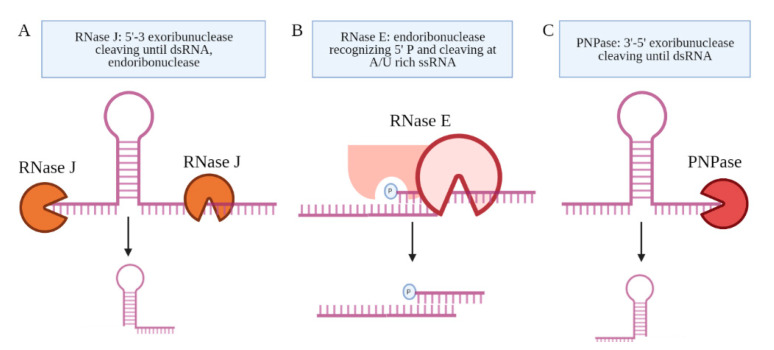
The proteins coupled to the Mtb degradosome are shown. (**A**) RNase J is an endoribonuclease and a 5′–3′ exoribonuclease. (**B**) RNase E is an endoribonuclease that cleaves ssRNA at A/U-rich sites after recognizing the 5′ phosphate in proximity. (**C**) PNPase is a 3′–5′ exoribonuclease, which is the only RNase implicated in the processing of an sRNA, Ms1.

**Table 1 ncrna-07-00069-t001:** List of sRNA identified by Arnvig et al., DiChiara et al., Gerrick et al., and Coskun et al.

Name	Northern or PCR Size	Location	Surrounding Genes	Expression
B11/6C (Candidate_1603) [8,9]	93	4099386-4099478 (−)	*rv3660c- rv3661*	H2O2 and pH = 5
B55 (Candidate_84) [8,9]	61	704187-704247 (+)	*rv0609A- rv0610c*	H2O2 and Mitomycin C
C8 (Mcr6, candidate_1621) [8,9,11]	58, 70, 128	4168154-4168281 (−)	*rv3722c- rv3723*	TBD ^a^
F6 (Mcr14, candidate_29) [8,9,25]	38, 58, 102	293604-293705 (+)	*fadA2-fadE5*	H2O2 and pH = 5
G2 (Candidate_1269) [8,9]	67, 214, 229	1914962-1915190 (−)	*tyrS-IprJ*	TBD
ASdes (candidate_121) [8,9,22]	48, 63, 68, 83, 94, 109, 149, 169, 195	918264-918458 (+)	within *desA1*	TBD
ASpks [9]	78, 89, 91, 102, 129, 142, 162	2299745-2299886 (+)	within *pks12*	H202
AS1726 [9]	61, 77, 85, 110, 213	1952291-1952503 (−)	within *Rv1726*	TBD
AS1890 [9]	63, 109, 191, 238	2139419-2139656 (+)	within *Rv1890*	TBD
MTS2823 or Ms1 [10,26,27]	250, 300	4100669-4100968 (+)	*rv3661- rv3662c*	in vivo
MTS1338/DrrS [10,28,29]	108, 109, ~160, 273	1960667-1960783 (+)	*rv1733c- rv1735c*	NO, stationary phase, in vivo
MTS0997/Mcr11 (Candidate_1693) [8,10,11,14,30]	115	1413094-1413224 (−)	*rv1264- rv1265*	in vivo, stationary phase, low pH, or hypoxia
Mcr1 [11]	>300	2029043-2029087 (TBD)	*ppe26-ppe27*	TBD
Mcr2 [11]	120	1108857-1108824 (TBD)	*rv0967- rv0968*	TBD
Mcr3 (candidate_190) [8,11]	118	1471619-1471742 (+)	*murA-rrs*	TBD
Mcr4 (candidate_1314) [8,11]	200–250	2137148-2137103 (TBD)	*fbpB-rv1887*	TBD
Mcr5 [11]	80	2437823-2437866 (−)	within *rv2175c*	TBD
Mcr7 [11,31]	350–400	2692172-2692521 (+)	*rv2395-pe_PGRS41*	TBD
Mcr8 (candidate_1935) [8,11]	200	4073966-4073908 (TBD)	*rv3661–rv3662c*	TBD
Mcr9 (candidate_1502) [8,11]	66–82	3317634-3317517 (TBD)	*ilvB1-cfp6*	TBD
Mcr10 [11]	120	1283693-1283815 (+)	within *rv1157c*	TBD
Mcr12 [11]	118	1228436-1228381 (TBD)	*rv1072- rv1073*	TBD
Mcr13 [11]	311	4315154-4315215 (TBD)	*rv3866- rv3867*	TBD
Mcr15 [11]	>300	1535417-1535716 (−)	*rv1363c- rv1364c*	TBD
Mcr16 [11]	100	2517032-2517134 (−)	within *fabD*	TBD
Mcr17 [11]	82–90	2905457-2905402 (TBD)	within *rv2613c*	TBD
Mcr18 [11]	82	3466287-3466332 (TBD)	within *nuoC*	TBD
Mcr19 [11]	66–82	575033-575069 (+)	within *rv0485*	TBD
ncRv11846/MrsI [12]	100	2096766-2096867 (+)	*blal-rv1847*	iron starvation, oxidative stress, and membrane stress
sncRNA-1 [13]	25	4352927-4352951	*esxA-rv3876*	inside macrophages
sncRNA-6 [13]	21	786003-786083	*rv0685- rv0686*	inside macrophages
sncRNA-8 [13]	24	1471701-1471724	*murA-rrs*	inside macrophages

^a^ TBD: to be determined.

**Table 2 ncrna-07-00069-t002:** List of Mtb-encoded proteins that have putative functions in sRNA processing.

Gene ID	Name	Species	Putative Function
*MTB000026*	RnpB	*M. bovis, Mtb, M. haemophilum*	RNA component of RNase P: RNase P catalyzes the removal of the 5′-leader sequence from pre-tRNA to produce the mature 5′ terminus.
*rv1340*	RphA	*M. marinum, M. leprae, M. bovis, Mtb*	Probable ribonuclease RphA (RNase PH).
*rv2092c*	HelY	*M. marinum, M. leprae, M. bovis, Mtb, M. abscessus*	DNA helicase activity.
*rv2179c*	Rnt	*Mtb, M. smegmatis, M. leprae, M. marinum*	Conserved hypothetical protein.
*rv2228c*	Rv2228c	*N/A*	Multifunctional protein. Has RNASE H, alpha-ribazole phosphatase, and acid phosphatase activities.
*rv2407*	Rnz	*Mtb, M. smegmatis, M. leprae, M. marinum, M. bovis*	Endonucleolytic cleavage of RNA, removing extra 3′ nucleotides from tRNA precursor, generating 3′ termini of tRNAs.
*rv2444c*	Rne	*M. bovis, Mtb, M. leprae, M. marinum, M. smegmatis*	Putative RNase E. Plays a central role in the maturation of 5S and 16S rRNAs and the majority of tRNAs. Also involved in the degradation of most mRNAs.
*rv2511*	Orn	*Mtb, M. smegmatis, M. leprae, M. marinum, M. bovis*	Involved in RNA degradation: 3′-to-5′ exoribonuclease specific for small oligoribonucleotides.
*rv2681*	Rnd	*Mtb, M. smegmatis, M. leprae, M. marinum, M. bovis*	Conserved hypothetical protein.
*rv2752c*	Rnj	*M. bovis, Mtb, M. leprae, M. marinum, M. smegmatis*	Conserved hypothetical protein.
*rv2783c*	GpsI (Pnp)	*M. marinum, M. leprae, M. bovis, Mtb, M. smegmatis, M. abscessus*	Involved in mRNA degradation. Hydrolyses single-stranded polyribonucleotides processively in the 3′ to 5′ direction.
*rv2902c*	RnhB	*M. marinum, M. leprae, M.bovis, Mtb, M. abscessus*	Probable ribonuclease HII protein RnhB.
*rv2907c*	RimM	*M. marinum, M. leprae, M. bovis, Mtb, M. smegmatis, M. abscessus*	Essential for efficient processing of 16S rRNA. Probably part of the 30S subunit prior to or during the final step in the processing of 16S free 30S ribosomal subunits. It could be some accessory protein needed for efficient assembly of the 30S subunit.
*rv2925c*	Rnc	*M. marinum, M. leprae, M. bovis, Mtb, M. smegmatis, M. abscessus*	Digests double-stranded RNA. Involved in the processing of ribosomal RNA precursors and of some mRNAs.
*rv3853*	RraA	*M. leprae, M. bovis, Mtb*	Regulator of RNase E activity a RraA.
*rv3923c*	RnpA	*M. marinum, M. leprae, M. bovis, Mtb, M. smegmatis, M. abscessus*	Ribonuclease P protein component RnpA.

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
