# Peer review of "Small RNAs Asserting Big Roles in Mycobacteria"

_ncrna, 2021, doi:10.3390/ncrna7040069_

Round 1

Reviewer 1 Report

I recommend to accept the manuscript.

Author Response

We thank the reviewer for their assessment of our article. The reviewer had accepted the prior sets of revisions.

Reviewer 2 Report

Comments to Authors

I have read the article entitled Small RNAs Asserting Big Roles in Mycobacteria. This article has gone through several rounds of review (I somehow counted two previous rounds), and thus, it is in a great state of completeness and accuracy. My understanding is that my main role here is to make sure that the previous comments from referees are properly addressed. From my own perspective, I cannot find major flaws, and I recommend to accept it after some minor concerns are addressed. The current review article comes in a great moment for the field of sRNAs, which is currently expanding. I thank authors for drafting such a complete overview of the non-coding RNA catalogue in Mtb. I find also relevant that authors have addressed as well the potential role of sRNAs in antibiotic resistance. Nice!

  • Comment 1

Regarding the previous comment about the 28% of Mtb transcriptome having a sRNA nature:

“L76-77: Not sure how the 28% was calculated

Answer: We have carefully re-read the manuscript from wherein our information was obtained. We have now corrected the information to accurately reflect what was reported. We also contacted Dr. Kristine Arnvig to confirm what her group had reported. See new lines 39-43.  Our new statement reads “In fact, ~28% of the Mtb transcriptome, characterized during exponential growth, contains non-coding RNA species, cis-encoded regulatory elements, antisense transcripts and intergenic sRNAs, that latter defined as > 100 nucleotides. The sRNAs make up 11% of the intergenic transcripts for all those with > 500 reads”. “

Despite the increased text, the main issue that I find, is that for the reader will not be easy to reproduce the calculation. The text keeps describing, but does not clearly states how was that calculated. This could become easily a number that will be reproduced in many future articles, and which no ones knows where it came from. I consulted also the article by Arnvig et al. and it is not easy to find the information. The authors of the current article have to think as well on the reader of the journal, especially on potential researchers interested in calculating how much of a transcriptome account for sRNAs.  That number was not exactly provided in the referred study, and it will definitely solve the issue if you could clearly state the numbers and the calculation. I think it is a good aspect that this review article provided. Until some extent is really surprising (and fascinating), that over a quarter of a transcriptome is comprised of sRNAs.

  • Comment 2

The case of MTS1338. I was reading the response from authors towards this issue. The presented modification of the text reflects what Arnvig et al. described in their study. I consider this matter corrected.

  • Comment 3

In line 121, I suggest to modify “selectively” for “selectivity”.

Author Response

We thank the reviewer for their comments and provide responses below each comment

Comment 1

Regarding the previous comment about the 28% of Mtb transcriptome having a sRNA nature:“L76-77: Not sure how the 28% was calculated. Answer: We have carefully re-read the manuscript from wherein our information was obtained. We have now corrected the information to accurately reflect what was reported. We also contacted Dr. Kristine Arnvig to confirm what her group had reported. See new lines 39-43.  Our new statement reads “In fact, ~28% of the Mtb transcriptome, characterized during exponential growth, contains non-coding RNA species, cis-encoded regulatory elements, antisense transcripts and intergenic sRNAs, that latter defined as > 100 nucleotides. The sRNAs make up 11% of the intergenic transcripts for all those with > 500 reads”. “

Despite the increased text, the main issue that I find, is that for the reader will not be easy to reproduce the calculation. The text keeps describing, but does not clearly states how was that calculated. This could become easily a number that will be reproduced in many future articles, and which no ones knows where it came from. I consulted also the article by Arnvig et al. and it is not easy to find the information. The authors of the current article have to think as well on the reader of the journal, especially on potential researchers interested in calculating how much of a transcriptome account for sRNAs.  That number was not exactly provided in the referred study, and it will definitely solve the issue if you could clearly state the numbers and the calculation. I think it is a good aspect that this review article provided. Until some extent is really surprising (and fascinating), that over a quarter of a transcriptome is comprised of sRNAs.

RESPONSE: We concur that this has been a difficult number to derive from the study in question. For this reason, we re-calculated the numbers using their Table 7 and supplemental Excel spreadsheet.  They compared the normalized read counts for sense RNAs (coding), antisense RNAs, and the IGR RNAs (total noncoding), excluding ribosomal RNAs. In some of their information, they considered read counts >500, while in other comments, read counts >5 were selected.  Based on our calculations from their data sets and a re-read of their abstract, the best estimates suggest a range from 22 to 28% Since they wrote in their published abstract twenty-five percent, we thought it best to simply write “approximately 25% of the of the transcriptome represents noncoding RNAs”.

Our changes are incorporated in lines lines 39-46. “First described as ranging in size from 50-350 nucleotides (nts) 8-12 these small RNAs now include some as small as 18 nts 13. The sRNAs, originally selected with sequences >100 nts in length, were found to represent ~11% of the intergenic transcripts (IGRs) identified from the exponential phase cultures. In addition to the sRNAs, IGRs include 5’ and 3’ UTRs, tRNAs and antisense RNAs. Based on the normalized read counts for sense, antisense, and intergenic noncoding RNAs, the antisense and intergenic noncoding RNAs made up roughly 25% of the transcripts mapping outside of ribosomal RNA genes 10

  • Comment 2

The case of MTS1338. I was reading the response from authors towards this issue. The presented modification of the text reflects what Arnvig et al. described in their study. I consider this matter corrected.

RESPONSE: No changes requested.

  • Comment 3

In line 121, I suggest to modify “selectively” for “selectivity”.

RESPONSE: Changes made in the text, now positioned at line 124. We thank the referee for this suggestion.

Reviewer 3 Report

The review is devoted to noncoding small RNAs in M. tuberculosis. The small RNAs described to date, their properties, possible mechanism of action, and their role in the pathogenesis of mycobacteria are discussed in detail. Despite the fact that such reviews are published quite regularly, the merits of this review include the consideration of a new class of small RNAs, sncRNAs, as well as modern view on sRNA processing. 

The authors have thoroughly analyzed the literature. However, for some unknown reasons, they completely ignored the small RNA Mcr7, for which the target was shown by Solans et al, 2014. Solans et al demonsrated that Mcr7 modulates translation of the tatC mRNA, and in this way influences TAT secretion apparatus. This information should be added and discussed in parts 3 “Functional Roles of Mycobacterial sRNAs…”  (Figure 2) and 4 “Regulation of Mycobacterial sRNAs/sncRNAs Expression” (Figure 3. It was shown that Mcr7 expression is under PhoP control)

The authors underlined the high homology of Ms1 of M.smegmatis and MTS2823 of M.tuberculosis.  However, the mechanism of Ms1 action was investigated for M.smegmatis only, and has not been proven for M.tuberculosis.  To be more accurate, it is better not to use generalization like “Ms1/MTS2823”, but separate the names.

Lines 42-44. References 8-12 refer to the statement “ranging in size”, ref 13 -  to the new class of sRNAs, sncRNAs. It is better to divide the references in the sentence.

Line 46.  Ref 14 describes Mcr11 only , but the phrase “…key roles in all aspects of mycobacterial growth and survival…” claims to be general. Add more references.

Line 62. sRNA-target mRNA interaction not only “changes translational efficiency”. Also, mRNA-sRNA complexes could be degraded by recruiting RNases, e.g. RNAse E. This phrase should be re-written in more general way

Lines 70-71. There are more recent works to illustrate this paragraph, e.g. Lu G et al  Two Small Extracellular Vesicle sRNAs Derived From Mycobacterium tuberculosis Serve as Diagnostic Biomarkers for Active Pulmonary Tuberculosis. Frontiers in microbiology. 2021;12. Please add to the references.

Table 1. Please explain the use of “candidate” in the column “Name”.  Add references for Mcr11

The column “Northern or PCR size” – please check the length of MTS1338 (Moores et al , 2017).

Author Response

We thank the reviewer for pointing out some key issues remaining. Our responses are as follows:

The review is devoted to noncoding small RNAs in M. tuberculosis. The small RNAs described to date, their properties, possible mechanism of action, and their role in the pathogenesis of mycobacteria are discussed in detail. Despite the fact that such reviews are published quite regularly, the merits of this review include the consideration of a new class of small RNAs, sncRNAs, as well as modern view on sRNA processing. 

The authors have thoroughly analyzed the literature. However, for some unknown reasons, they completely ignored the small RNA Mcr7, for which the target was shown by Solans et al, 2014. Solans et al demonstrated that Mcr7 modulates translation of the tatC mRNA, and in this way influences TAT secretion apparatus. This information should be added and discussed in parts 3 “Functional Roles of Mycobacterial sRNAs…”  (Figure 2) and 4 “Regulation of Mycobacterial sRNAs/sncRNAs Expression” (Figure 3. It was shown that Mcr7 expression is under PhoP control)

RESPONSE: We apologize for having missed this particular article. We have now incorporated into multiple sections in the article. Please see in Figures 1, 2 and lines 157, 169, 243, 272 and 275. Findings from Solans et al paper were added to relevant sections.

The authors underlined the high homology of Ms1 of M. smegmatis and MTS2823 of M. tuberculosis.  However, the mechanism of Ms1 action was investigated for M. smegmatis only and has not been proven for M. tuberculosis.  To be more accurate, it is better not to use generalization like “Ms1/MTS2823” but separate the names.

RESPONSE: We have separated the names as suggested to Ms1.

Lines 42-44. References 8-12 refer to the statement “ranging in size”, ref 13 -  to the new class of sRNAs, sncRNAs. It is better to divide the references in the sentence.

RESPONSE: We have separated the references as requested and this is seen in lines 39-40.

Line 46.  Ref 14 describes Mcr11 only, but the phrase “…key roles in all aspects of mycobacterial growth and survival…” claims to be general. Add more references.

RESPONSE: In line 48, we have added more references.

Line 62. sRNA-target mRNA interaction not only “changes translational efficiency”. Also, mRNA-sRNA complexes could be degraded by recruiting RNases, e.g. RNAse E. This phrase should be re-written in more general way.

RESPONSE: In lines 64-65, we made these changes. “This interaction changes the translation efficiency of the corresponding gene or induce the degradation of the target mRNA 20, 21. “

Lines 70-71. There are more recent works to illustrate this paragraph, e.g. Lu G et al  Two Small Extracellular Vesicle sRNAs Derived From Mycobacterium tuberculosis Serve as Diagnostic Biomarkers for Active Pulmonary Tuberculosis. Frontiers in microbiology. 2021;12. Please add to the references.

RESPONSE: We have added this reference and thank the reviewer as we did not notice this particular article in our earlier versions.

Table 1. Please explain the use of “candidate” in the column “Name”.  Add references for Mcr11

RESPONSE: We apologize for the confusion on the use of ‘Candidate’. Candidate_Number refers to the terms derived from Pellin et al. 2012 paper that used a bioinformatic approach to identify 1948 candidate small RNAs. For those which were identified previously, the names used in different studies were pooled together in the table.

For MTS0997/Mcr11, there are 2 papers elucidating the function of this sRNA by Girardin et al. in 2018 and 2020 which are already cited. The other 3 papers cited are those for which identified this sRNA. We are not aware of specific paper that needs to be.

The column “Northern or PCR size” – please check the length of MTS1338 (Moores et al , 2017).

RESPONSE: We have done so and corrected the information in the Table. Two sRNA with alternative 3’ ends yielding sRNAs at 108 and 109 nts, an intermediate sized-sRNA ~160 nts, and a putative precursor around 273 nts are now presented.

Reviewer 4 Report

Mycobacterium tuberculosis (Mtb), the cause of Tuberculosis (TB), leads among human infectious diseases with high mortality. Prior studies have identified various small RNAs from Mtb culture, infected cells and patients, including 50-350nt sRNAs and <50nt sncRNAs. As the experimental tools and high throughput sequencing became more accessible, more studies started switching to characterize the functions of Mtb-encoded sRNAs/sncRNAs.

In this review, the authors provide a very comprehensive survey of small RNA studies in mycobacteria. They reviewed the identification, biogenesis/processing, and protein partners/chaperons of Mtb sRNAs/sncRNAs.

Overall, the review is written and organized well. It provides valuable information on the current status and future directions of small RNAs research in Mtb. However, there is a limitation to the review. The authors spent most energy discussing small RNAs in Mtb itself. But as a pathogen, it would be equally important to talk about the disease and its interaction with the host. This review will be significantly enhanced if additional insights are provided from this aspect, even if most studies are still ongoing.

Specific comments:

  1. Regarding the biogenesis of Mtb sRNAs/sncRNAs, what type of precursors they derived from, dsRNAs or ssRNAs?

  1. Since Mtb is the cause of TB, it would be very interesting to review the current progress in the field from the disease perspective. For example, whether and how sRNAs/sncRNAs affect the pathogenicity of Mtb.

  1. I was wondering, is there any study comparing the Mtb sRNA/sncRNAs repertoire between culture and infected cells/patients (which is without/with the host)? If yes, are they different?

  1. As the authors pointed out in the review, some sRNAs/sncRNAs were detectible in the plasm of TB patients. I'm curious whether Mtb-encoded sRNAs/sncRNAs play any roles in the interactions between the pathogen and the host. For example, does Mtb-encoded sRNAs potentially target any genes in the human genome? Is there any machinery from the host (e.g. transcription, RNA processing, etc.) required for the biogenesis and processing of Mtb sRNAs/sncRNAs?

Minor:

  1. Figure 2A and text: the authors stated that B11/6C activates two genes panD and dnaB, which promotes growth. However, that seems to contradict the statement that overexpression of B11/6C results in growth inhibition. Here need some clarifications.

Author Response

We thank the reviewer for taking the time to assess our review article, which has been revised several times already. We provide the following responses to address the comments/concerns they noted.

Overall, the review is written and organized well. It provides valuable information on the current status and future directions of small RNAs research in Mtb. However, there is a limitation to the review. The authors spent most energy discussing small RNAs in Mtb itself. But as a pathogen, it would be equally important to talk about the disease and its interaction with the host. This review will be significantly enhanced if additional insights are provided from this aspect, even if most studies are still ongoing.

Specific comments:

1. Regarding the biogenesis of Mtb sRNAs/sncRNAs, what type of precursors they derived from, dsRNAs or ssRNAs?

RESPONSE: We have described in line 320 that the precursors of sRNAs are indicated as ssRNAs. However, we cannot rule out that some sRNAs form secondary structures involving dsRNA with stem loops. sncRNAs were shown to derive from such secondary structures and this is mentioned in lines 329-330.      

2. Since Mtb is the cause of TB, it would be very interesting to review the current progress in the field from the disease perspective. For example, whether and how sRNAs/sncRNAs affect the pathogenicity of Mtb.

RESPONSE: Mycobacterial sRNAs are relatively new in the field (1st identified in 2009 by Arnvig et al.). To date, they have been studied in the in vitro cultures and we clearly need more studies to understand their functions in vivo. sncRNAs were identified in infected cells and among them it was shown that sncRNA-1 overexpression increased the bacterial burden in macrophages (Coskun et al. 2020) indicating a role in pathogenesis. This remains an important area of research that is being pursued by many groups.

3. I was wondering, is there any study comparing the Mtb sRNA/sncRNAs repertoire between culture and infected cells/patients (which is without/with the host)? If yes, are they different?

RESPONSE: Fu et al. 2018 paper has compared mycobacterial small RNA in the bacterial culture supernatant and plasma of patients with active tuberculosis. Only ASdes was detected in the plasma of the patients. Very recently, Han et al. 2021 have screened 20 sRNAs in the plasma of TB patients for detection of active pulmonary tuberculosis and they were able to detect and found MTS2823, MTS0997, MTS1338, ASdes, G2, C8, mcr15 and MTS1082 in at least 50% of the patients. Most studies involving patient sera investigate the use of such sRNAs as biomarkers. Therefore, mechanistic studies are still lacking in the field.  

Unlike sRNAs, smaller RNAs have not been studied in free floating mycobacteria, but identified in the host. Mtb-encoded 35 sncRNAs were identified in infected cells by Coskun et al. 2020. Two additional studies have identified miRNA-like sRNAs: the 1st by Furuse et al. in 2014 identifying a single RNA in infected cells and the 2nd identifying 6 RNAs from the serum of the patients with TB.

4. As the authors pointed out in the review, some sRNAs/sncRNAs were detectible in the plasm of TB patients. I'm curious whether Mtb-encoded sRNAs/sncRNAs play any roles in the interactions between the pathogen and the host. For example, does Mtb-encoded sRNAs potentially target any genes in the human genome? Is there any machinery from the host (e.g. transcription, RNA processing, etc.) required for the biogenesis and processing of Mtb sRNAs/sncRNAs?

RESPONSE: Most of the sRNA in mycobacteria were initially identified from the in vitro cultures, which implies that they do not require a host machinery to be produced. However, some were identified in the serum of patients with TB such as ASdes (Fu et al. 2018 and Lu et al. 2021). This suggests that they are either secreted or released with the lysis of the bacteria to the host’s blood stream. It is possible that they have regulatory functions in the host. However, this is a newly emerging field and currently there is no study that have addressed this issue to our knowledge. sRNAs identified in the blood stream were identified to be used as biomarkers.  Two studies have identified miRNA-like sRNAs: the 1st by Furuse et al. in 2014 identifying a single RNA in infected cells. This miRNA-like RNA uses host’s miRNA processing machinery for its processing, but its targets remain unknown (please see lines 108-113). And in the 2nd study identifying 6 RNAs from the serum of the patients with TB, functional studies with these miRNA-like RNAs has not been reported (please see lines 114-116). A final comment is that the sncRNAs were identified in infected cells. sncRNA-1 and sncRNA-6 were solely detected in infected cells, whereas sncRNA-8 was detected independent of an infection yet induced in infected cells (Coskun et al. 2020). However, “mature” sncRNAs can be picked up in vitro in clones overexpressing the sncRNAs. This implies that the sncRNAs can be processed in mycobacteria independent of an infection and their processing does not require a host component. Among the sncRNAs identified, it was shown that sncRNA-1 overexpression increased the bacterial burden in macrophages (Coskun et al. 2020) indicating a role in pathogenesis, but this is thought to be mediated by transcriptome changes within mycobacteria. We cannot rule out that some sncRNA are released to the host and regulate the host transcriptome, but this is to be explored in the future.      

Minor:

  1. Figure 2A and text: the authors stated that B11/6C activates two genes panD and dnaB, which promotes growth. However, that seems to contradict the statement that overexpression of B11/6C results in growth inhibition. Here need some clarifications.

RESPONSE: We apologize for the confusion. Between lines 189-194, we clarify this by stating that B11 regulates genes coupled to DNA replication and protein secretion. This was done with bioinformatic searches which did not reveal if this is a positive or negative regulation. Later mechanistic studies with the predicted targets revealed that this is a negative regulation.

This manuscript is a resubmission of an earlier submission. The following is a list of the peer review reports and author responses from that submission.

Round 1

Reviewer 1 Report

Overall, the structure of the paper is confusing with same sRNAs mentioned in several sections. The language is unclear in many places. Perhaps a different structure would make it more accessible.

There are several incorrect statements and seriously wrong terminology (e.g. confusing promoters, UTRs and transcription start sites really is an undergrad mistake). 

Specific comments.

L 75 The increase in sRNA expression during log phase growth is incorrect

L 173-75 There is no mention of MTS1338 overexpression inhibiting Mtb replication in ref. 8

L232 mycobacterial SigA only requires the -10 hexamer

L 237-8 Authors mix up promoter with 5’ UTR

L 238 sRNAs do not have a 3’ UTR

L 239 specify 5’ and 3’ elements!

L 256-59 If the TSS for Mcr11 overlaps with the -35 hexamer of AmbR, these transcripts will not overlap and they will not share a TSS.

L 259-67 There is no suggestion of shared TSS between Rv1733c and DrrS; in fact the TSS are mapped (Cortes et al. 2013) and are ~190 bp apart

But looking at the figure (3) it seems the problem is confused terminology, i.e. the definition of ‘transcription start site’ versus ‘promoter’

L 329-331 why would tRNAs be ‘processed’ by polyA polymerase?

L 389 what do the authors mean by type of regulatory elements controlling sRNA transcription? Do you mean to say that there are sRNA-specific promoter elements? I find that unlikely. And there are plenty of sRNAs identified, it is the further characterisation that is missing.

L 395-96 RNA editing and processing seems to be bunged into same category here 

Author Response

Reviewer 1:

Overall, the structure of the paper is confusing with same sRNAs mentioned in several sections. The language is unclear in many places. Perhaps a different structure would make it more accessible.

There are several incorrect statements and seriously wrong terminology (e.g. confusing promoters, UTRs and transcription start sites really is an undergrad mistake). 

Response: We have rearranged some of the sections to remove the redundancy. We have also corrected many of the statements and fixed the terminology. The revisions were extensive, with changes incorporated throughout the manuscript.

Specific comments.

L 75 The increase in sRNA expression during log phase growth is incorrect

Response: Corrected

L 173-75 There is no mention of MTS1338 overexpression inhibiting Mtb replication in ref. 8

Response: Corrected

L232 mycobacterial SigA only requires the -10 hexamer

L 237-8 Authors mix up promoter with 5’ UTR

Response: Corrected

L 238 sRNAs do not have a 3’ UTR

Response: Corrected

L 239 specify 5’ and 3’ elements!

Response: Corrected

L 256-59 If the TSS for Mcr11 overlaps with the -35 hexamer of AmbR, these transcripts will not overlap and they will not share a TSS.

Response: corrected

L 259-67 There is no suggestion of shared TSS between Rv1733c and DrrS; in fact the TSS are mapped (Cortes et al. 2013) and are ~190 bp apart

But looking at the figure (3) it seems the problem is confused terminology, i.e. the definition of ‘transcription start site’ versus ‘promoter’

Response: Corrected

L 329-331 why would tRNAs be ‘processed’ by polyA polymerase?

Response: CCA-adding enzymes and poly(A) polymerases are both members of the same nucleotidyltransferase superfamily and there is only one member of this family encoded by M. tuberculosis – Rv3907c. While it is annotated as Poly(A) polymerase (and thus the name in the manuscript is kept as Poly(A) polymerase) it rather plays a role in addition of CCA, same as in B. subtilis. Nonetheless, these enzymes are known to play pleiotropic roles in the cells, thus other activities cannot be ruled out.

L 389 what do the authors mean by type of regulatory elements controlling sRNA transcription? Do you mean to say that there are sRNA-specific promoter elements? I find that unlikely. And there are plenty of sRNAs identified, it is the further characterisation that is missing.

Response: Clarified.

L 395-96 RNA editing and processing seems to be bunged into same category here 

Response: Clarified

Reviewer 2 Report

This review provide important basis for future studies on the Mycobacteris and tuberculosis.

Here I suggest that to discuss supplementarily abourt how antibiotics induce small RNA expression and how small RNA regulate the M.tb drug resistance in the review?

Author Response

This review provide important basis for future studies on the Mycobacteris and tuberculosis.

Here I suggest that to discuss supplementarily abourt how antibiotics induce small RNA expression and how small RNA regulate the M.tb drug resistance in the review?

Response: Included as suggested.

Reviewer 3 Report

The review focuses on small regulatory RNAs and proteins playing roles in their expression, processing and functions. Interestingly, the review also deals with small non-coding RNAs (sncRNAs) and mentions experimental approach to study sncRNAs functions in bacteria.

In the first part of the review, the authors provide the list of M. tuberculosis sRNAs and discuss their functions. However, some statements are incorrect and there are also mistakes in Figures (see points 1, 3, 6). In addition, some parts of the review are very speculative – for example the paragraph „tRNA processing enzymes as potential players for sRNA maturation“ includes description of tRNA processing but provides no evidence that tRNA processing enzymes could play a role in processing of sRNAs. I would expect some indications from other bacteria or eukaryotes that led the authors to the conclusion that tRNA processing enzymes are potential players in sRNA maturation in mycobacteria, but this information is missing. Similarly, in the „Hunt for the mycobacterial Hfq equivalent“ paragraph, an important paper from Mai, NAR, 2019 (Mycobacterium tuberculosis 6C sRNA binds multiple mRNA targets via C-rich loops independent of RNA chaperones) is not discussed. This article shows that no RNA chaperone is needed for 6C RNA base-pairing and mRNA target regulation and suggests that no such RNA chaperon probably exists in GC-rich mycobacteria at all. This possibility should be discussed.

Taken together, this is a useful and interesting review but changes are clearly required to make it more accurate and reader-friendly (e.g. point 4). It would be helpful to better explain key experiments mentioned in the manuscript as some parts are quite difficult to understand (points 5, 8). Specific comments are below.

  1. Line 74-75, the statement „MTS0997 and MTS2823 are detected in stationary cultures and increase 12-fold and 1.6-fold during exponential phase“ is incorrect. The authors refer to Arnvig et al., 2009 (Sequence-Based Analysis Uncovers an Abundance of Non-Coding RNA in the Total Transcriptome of Mycobacterium tuberculosis), but in this article it is shown that the expression of MTS0997 and MTS2823 actually INCREASES in stationary phase; and not in exponential phase. MTS2823 is one of the most abundant transcripts in stationary phase!
  2. In Table 1, column Expression. What do the abbreviations „TBD“ and „MMC“ mean? All abbreviations should be explained.
  3. Figure 1 should be deleted as it is confusing/incorrect. For example, the predicted structure of MrsI in this figure differs from what was published previously (Gerrick et al., 2018, Small RNA profiling in Mycobacterium tuberculosis identifies MrsI as necessary for an anticipatory iron sparing response). Why? The seed region of MrsI in Fig. 1 is hidden in the helix, while in the previously published article it is in the loop which is probably correct as the seed regions were shown to be mainly in loops in mycobacteria. Also, the MTS2823/Ms1 structure differs from predicted structure published before (Hnilicova et al., 2014, Ms1, a novel sRNA interacting with the RNA polymerase core in mycobacteria). This figure is therefore confusing and does not correspond to the published data.
  4. Based on the paragraph composed of lines 91-115 I would conclude that sncRNAs are generated by eukaryotic miRNA processing pathways from bacterial transcripts, therefore they are present only in intracellular pathogens. Is that true and could the authors better explain if sncRNAs are present also in in vitro growing M. tuberculosis?
  5. Could the authors also better explain some of the findings – e.g. „In a broader screen for miRNA-like sequences using additional selection criteria that included coverage of plant small RNAs, a set of 35 small RNAs were identified in Mtb-infected macrophages“ (line 108). What are „plant small RNAs criteria“?
  6. Figure 2C shows that MTS2823/Ms1 binds to RNA polymerase-SigA complex. This is clearly NOT true. In Hnilicova et al. 2014 (Ms1, a novel sRNA interacting with the RNA polymerase core in mycobacteria) the authors show that Ms1 interacts only with RNA polymerase and not with SigA, which differentiates Ms1 from 6S RNA. Please correct the figure. Also, what is the small almost invisible yellow thing at the end of the MTS0997 pathway (Fig. 2D)? Fragmented RNA? Fatty acids?
  7. Lines 287-288: „Both sncRNA-1 and sncRNA-6, which have the final sizes of 25 nts and 21 nts, require processing enzymes for their generation (Figure 3).“ But in Fig. 3 sncRNAs are not shown at all (?).
  8. The description of the experiment mentioned in lines 159-162 is rather complicated and I feel that the experiment is so important that should be discussed in the section dedicated to Hfq.
  9. The text contains many missing or additional spaces, e.g. line 110 „from18-30 nts“, line 189 „11 nucleotide    region“, line 257 and especially Table 2 - e.g. „mature5´terminus“ or „thedegradation“.
  10. Citations are missing: line 88 „Such studies..“ Which studies? There is only one citation (Gerrick et al., 2018) referring to sRNAs identification in stress condition in the previous sentence. Line 245. Legends of the Fig. 2, 3 and 4.

Author Response

The review focuses on small regulatory RNAs and proteins playing roles in their expression, processing and functions. Interestingly, the review also deals with small non-coding RNAs (sncRNAs) and mentions experimental approach to study sncRNAs functions in bacteria.

In the first part of the review, the authors provide the list of M. tuberculosis sRNAs and discuss their functions. However, some statements are incorrect and there are also mistakes in Figures (see points 1, 3, 6). In addition, some parts of the review are very speculative – for example the paragraph „tRNA processing enzymes as potential players for sRNA maturation“ includes description of tRNA processing but provides no evidence that tRNA processing enzymes could play a role in processing of sRNAs. I would expect some indications from other bacteria or eukaryotes that led the authors to the conclusion that tRNA processing enzymes are potential players in sRNA maturation in mycobacteria, but this information is missing. Similarly, in the „Hunt for the mycobacterial Hfq equivalent“ paragraph, an important paper from Mai, NAR, 2019 (Mycobacterium tuberculosis 6C sRNA binds multiple mRNA targets via C-rich loops independent of RNA chaperones) is not discussed. This article shows that no RNA chaperone is needed for 6C RNA base-pairing and mRNA target regulation and suggests that no such RNA chaperon probably exists in GC-rich mycobacteria at all. This possibility should be discussed.

Taken together, this is a useful and interesting review but changes are clearly required to make it more accurate and reader-friendly (e.g. point 4). It would be helpful to better explain key experiments mentioned in the manuscript as some parts are quite difficult to understand (points 5, 8). Specific comments are below.

Response: We have made corrections throughout the manuscript to address the concerns raised by the reviewer. The suggestions from the reviewer have allowed us to prepare a more precise the clear review article.

  1. Line 74-75, the statement „MTS0997 and MTS2823 are detected in stationary cultures and increase 12-fold and 1.6-fold during exponential phase“ is incorrect. The authors refer to Arnvig et al., 2009 (Sequence-Based Analysis Uncovers an Abundance of Non-Coding RNA in the Total Transcriptome of Mycobacterium tuberculosis), but in this article it is shown that the expression of MTS0997 and MTS2823 actually INCREASES in stationary phase; and not in exponential phase. MTS2823 is one of the most abundant transcripts in stationary phase!

Response: Corrected

  1. In Table 1, column Expression. What do the abbreviations „TBD“ and „MMC“ mean? All abbreviations should be explained.

Response: Corrected

  1. Figure 1 should be deleted as it is confusing/incorrect. For example, the predicted structure of MrsI in this figure differs from what was published previously (Gerrick et al., 2018, Small RNA profiling in Mycobacterium tuberculosis identifies MrsI as necessary for an anticipatory iron sparing response). Why? The seed region of MrsI in Fig. 1 is hidden in the helix, while in the previously published article it is in the loop which is probably correct as the seed regions were shown to be mainly in loops in mycobacteria. Also, the MTS2823/Ms1 structure differs from predicted structure published before (Hnilicova et al., 2014, Ms1, a novel sRNA interacting with the RNA polymerase core in mycobacteria). This figure is therefore confusing and does not correspond to the published data.

Response: Corrected

  1. Based on the paragraph composed of lines 91-115 I would conclude that sncRNAs are generated by eukaryotic miRNA processing pathways from bacterial transcripts, therefore they are present only in intracellular pathogens. Is that true and could the authors better explain if sncRNAs are present also in in vitro growing M. tuberculosis?

Response: Clarified

  1. Could the authors also better explain some of the findings – e.g. „In a broader screen for miRNA-like sequences using additional selection criteria that included coverage of plant small RNAs, a set of 35 small RNAs were identified in Mtb-infected macrophages“ (line 108). What are „plant small RNAs criteria“?

Response: Clarified

  1. Figure 2C shows that MTS2823/Ms1 binds to RNA polymerase-SigA complex. This is clearly NOT true. In Hnilicova et al. 2014 (Ms1, a novel sRNA interacting with the RNA polymerase core in mycobacteria) the authors show that Ms1 interacts only with RNA polymerase and not with SigA, which differentiates Ms1 from 6S RNA. Please correct the figure. Also, what is the small almost invisible yellow thing at the end of the MTS0997 pathway (Fig. 2D)? Fragmented RNA? Fatty acids?

Response: Corrected

  1. Lines 287-288: „Both sncRNA-1 and sncRNA-6, which have the final sizes of 25 nts and 21 nts, require processing enzymes for their generation (Figure 3).“ But in Fig. 3 sncRNAs are not shown at all (?).

Response: Corrected

  1. The description of the experiment mentioned in lines 159-162 is rather complicated and I feel that the experiment is so important that should be discussed in the section dedicated to Hfq.

Response: Changed accordingly

  1. The text contains many missing or additional spaces, e.g. line 110 „from18-30 nts“, line 189 „11 nucleotide    region“, line 257 and especially Table 2 - e.g. „mature5´terminus“ or „thedegradation“.

Response: Corrected

  1. Citations are missing: line 88 „Such studies..“ Which studies? There is only one citation (Gerrick et al., 2018) referring to sRNAs identification in stress condition in the previous sentence. Line 245. Legends of the Fig. 2, 3 and 4.

Response: Corrected

Round 2

Reviewer 1 Report

I had a closer look, but frankly I still think it is very poor.

Below I have pasted some of the preliminary findings. You will see that some things have not been addressed in spite of the authors claiming they have been and there are still some misunderstandings in very basic terminology.   

L 55-57: sRNAs transcribed from the UTR of a coding gene are not AS-RNAs 

L 59: whether the interaction inhibits or enhances translation would depend on where the interaction occurs 

L76-77: Not sure how the 28% was calculated 

L 131-132: why is it unlikely that they are generated by host processing enzymes? What is known about their AU preference? Is the rationale that the miRNA-like sRNAs are generated by different pathways in different pathogens (i.e. Salmonella vs Mtb) depending on the GC content? 

L 228: there is still no mention of MTS1338 overexpression in the cited paper (this was flagged in previous report and claimed to be corrected) 

L 232: DGE associated with MTS2823 overexpression was analysed by microarrays not RNA-seq

L280: do the authors mean transform instead of transduce (phage)?  

L 313: there are no factor dependent terminators in the Miotto paper 

L 314: what is meant by 5’ and 3’ elements? Specify 

L 316: what is meant by ‘neither site’? 

L 333-334: they do not share a promoter; how would ‘sharing a promoter’ work with respect to -35 and -10? 

L343-344: also mentioned in previous report, transcripts from opposite strands do not share a transcription start site 

Author Response

Reviewer 1: We thank reviewer 1 for their detailed assessment of our review article. We concur with all their suggestions and respond as follows. Please note the line numbering does not match with the new submission due to formatting changes.

L 55-57: sRNAs transcribed from the UTR of a coding gene are not AS-RNAs 

Answer: We rephrased this sentence to read ‘The sRNAs transcribed in the antisense orientation to a coding gene or to the untranslated region of the gene are called antisense RNAs.’

L 59: whether the interaction inhibits or enhances translation would depend on where the interaction occurs 

Answer: Corrected as ‘This interaction changes the translation efficiency of the corresponding gene.’ 

L76-77: Not sure how the 28% was calculated 

Answer: This was reported by Arnvig et. al. 2011 in the Discussion section.

L 131-132: why is it unlikely that they are generated by host processing enzymes? What is known about their AU preference? Is the rationale that the miRNA-like sRNAs are generated by different pathways in different pathogens (i.e. Salmonella vs Mtb) depending on the GC content? 

Answer: We updated this to read ‘This suggests these small RNAs are unlikely to be generated by eukaryotic miRNA processing enzymes as miRNAs have GC contents closer to 50% 30.’

L 228: there is still no mention of MTS1338 overexpression in the cited paper (this was flagged in previous report and claimed to be corrected) 

Answer: We re-checked this reference and confirmed the correct location of the Ignatov manuscript.

L 232: DGE associated with MTS2823 overexpression was analysed by microarrays not RNA-seq

Answer: We thank the reviewer for identifying this oversight. We now added ‘High levels of MTS2823 (300 nts) also inhibit Mtb growth, with transcriptome analysis using microarrays revealing many transcripts involved in metabolism that are downregulated (Figure 2B, Table 1).’

L280: do the authors mean transform instead of transduce (phage)?

Answer: Corrected as ‘These LNA-PIs were tested in mycobacteria, which are inherently difficult to transform.’  

L 313: there are no factor dependent terminators in the Miotto paper 

Answer: We have adjusted our sentence to read ‘Of the sRNAs identified in the screen, 46.9% had the consensus SigA promoter sequence in the upstream of the 5’ end, with 8.5% containing an intrinsic or factor-independent terminator sequence in the downstream or 3’ end.’

L 314: what is meant by 5’ and 3’ elements? Specify 

Answer: We explain, in more detail both the SigA binding sites and termination sequences.

L 316: what is meant by ‘neither site’? 

Answer: We corrected this to state that in certain sRNAs, the SigA and termination sequences were not found

L 333-334: they do not share a promoter; how would ‘sharing a promoter’ work with respect to -35 and -10? 

Answer: This was somewhat confusing and not well-delineated, so we removed this sentence from the article.

L343-344: also mentioned in previous report, transcripts from opposite strands do not share a transcription start site 

Answer: see above response as the sentence was deleted

Reviewer 3 Report

I have only minor suggestions to improve the comprehensibility of the manuscript, otherwise I recommend to accept the manuscript:

  1. Could the authors use the same format of the references in Table 1 and in the main text? In the main text, the references are numbered and in the Table, the references are as name, year. For example the sentence „Thirty-seven sRNAs have been identified….confirmed by Northern blotting (Table 1)11“, line 102 – it is not clear to the readers which sRNAs in the Table 1 are discussed.
  2. The authors changed the sRNA structures in Fig. 1, could the authors use the same sRNA structures also in the Fig. 2, if possible? It is a bit confusing for the readers that structures of the same sRNA in Fig. 1 and Fig. 2 are different. 
  3. Line 381: „The indicated that..“ should be probably „This indicated..“
  4. Please add citations to the sentences at line 59 and lines 62-63.

Author Response

Reviewer 3. We thank the 3rd reviewer for their helpful and constructive criticisms. Enclosed please find all the changes we made based on their queries.

I have only minor suggestions to improve the comprehensibility of the manuscript, otherwise I recommend to accept the manuscript:

  1. Could the authors use the same format of the references in Table 1 and in the main text? In the main text, the references are numbered and in the Table, the references are as name, year. For example the sentence „Thirty-seven sRNAs have been identified….confirmed by Northern blotting (Table 1)11“, line 102 – it is not clear to the readers which sRNAs in the Table 1 are discussed.

Answer: Changed the references in the table to numbers.

  1. The authors changed the sRNA structures in Fig. 1, could the authors use the same sRNA structures also in the Fig. 2, if possible? It is a bit confusing for the readers that structures of the same sRNA in Fig. 1 and Fig. 2 are different. 

Answer: Updated figure 2 accordingly.

  1. Line 381: „The indicated that..“ should be probably „This indicated..“

Answer: Corrected.

  1. Please add citations to the sentences at line 59 and lines 62-63.

Answer: Citations inserted.